# Development of Method Enhanced Laser Ablation Efficiency According to Fine Curvature of the Polymer through the Preliminary Preparation Process Using UV Picosecond Laser

**DOI:** 10.3390/polym12040959

**Published:** 2020-04-20

**Authors:** Seung Sik Ham, Ho Lee

**Affiliations:** 1Institute for Nanophotonics Applications, Kyungpook National University, 80, Daehak-ro, Buk-gu, Daegu 41566, Korea; ssh13@knu.ac.kr; 2School of Mechanical Engineering, Kyungpook National University, 80, Daehak-ro, Buk-gu, Daegu 41566, Korea; 3Laser Application Center, Kyungpook National University, 70, Dongnae-ro, Dong-gu, Daegu 41061, Korea

**Keywords:** picosecond pulse laser, fine curvature of polymer, enhanced laser ablation efficiency, artificial chimney, laser ablation

## Abstract

In processes using the ultrashort pulsed laser, the phenomenon that the ablation efficiency is reduced due to the increase of the shielding effect of the generated plume is increasingly caused by the use of the high power and high repetition rate. A new method is needed to prevent a decrease in ablation efficiency in processing using an ultrashort pulsed laser. In this study, the proposed a processing method that can improve the ablation efficiency by providing an efficient escape path of plume, and examine the feasibility of a new processing method. The new method we proposed is a method of laser processing after generating a fine curvature in the polymer as a preliminary preparation. The fine curvature of the polymer produced by the preliminary preparation induces an artificial chimney-like opening along the path of the incident beam during laser processing, thereby enabling the plume to be effectively removed. The experiment for examine the feasibility through a new method was conducted using a 10-picosecond laser of UV wavelength with two optical systems. As a new processing method, when processing with ultrashort pulse laser, it was observed that the ablation efficiency improved.

## 1. Introduction

Flexible devices, such as flexible displays [1,2,3] and batteries [4,5,6], are drawing attention. For flexible devices, flexible materials such as polymers are used instead of conventional hard materials such as glasses. In general, the requirements of each application should be satisfied in order to utilize flexible polymers in the applications of flexible devices. Active research on laser ablation methods is being conducted as a solution for precision machining that is appropriate these requirements [7,8,9].

The first polymer processing using laser was performed in 1982 by Srinvasan et al. [10] and Kawamura et al. [11]. Since then, many researchers have conducted studies on laser processing using various polymers used in a variety of applications. In general, ultraviolet (UV) laser is used for laser processing of polymer materials because it can process materials by linear absorption. UV laser is characterized by much larger photon energy than that of the laser of infrared wavelength range. When laser processing is performed using large photon energy of the UV wavelength, the photochemical mechanism has a dominant effect. Thus, when the UV laser processing result is compared with the result of laser processing of different wavelengths, the UV laser processing result generates less thermal damage of the periphery and smaller secondary damage. In other words, UV laser processing has the advantage of more precise processing than other wavelengths. Owing to these advantages, the UV laser is used in PCB or F-PCB processing [8,12,13,14], lithography [15,16,17,18], ITO patterning [19,20,21], 3D printing [22,23], and other applications.

However, in most cases, UV lasers are inappropriate for existing machining methods—such as cutting and welding processes—because it has a low average power that those of lasers of other wavelengths. Furthermore, excimer laser using a gas as laser medium is difficult to manage although the beam quality is good [24], whereas excimer laser using a solid material as laser medium has a poorer beam quality than that of excimer laser [11,16,17]. Furthermore, special safety measures are required when using laser of UV wavelengths because they are more harmful to human body than other wavelengths.

Laser processing using ultrashort pulse lasers, which began in earnest in the 1990s as a measure to overcome the limitations of UV laser, attracted much attention for the following advantages. First, it enables “cold ablation” which is a type of processing with almost no thermal damage to the periphery of laser irradiated part and almost no secondary damage [25,26,27]. Second, it enables laser processing by the optical phenomenon of nonlinear absorption even with wavelengths at which materials to be processed are not linearly absorbed [28,29,30]. This means that it is possible to process materials without depending on wavelength.

The reason that ultrashort pulse laser processing enables “cold ablation” is because materials are processed with a pulse width (picoseconds–femtoseconds) shorter than the thermal relaxation time (for example, several tens of picoseconds for metals), which is the unique characteristic of materials. In other words, it is possible because processing is completed before the laser is irradiated and the heat generated by the interaction between laser and material is transmitted to the surroundings. Thus, due to the possibility of cold ablation, ultrashort pulse laser has been used widely as a high-precision two- and three-dimensional processing method for various materials (e.g., glass, metal, polymer).

At first, the performance of ultrashort pulse laser was several watts output and several kHz pulse repetition rate, but with the recent development of technology, an output of several hundred watts and repetition rates of several to several tens MHz or even to several GHz have become possible. As a result, completely different processing patterns have appeared. Many researchers predicted that when the technology of ultrashort pulse laser is further developed to high power and high repetition rate, it would only increase the ablation efficiency while enabling ‘cold ablation’ processing with low thermal and secondary damages around the periphery of laser irradiation.

However, even though the technology has developed to high power and high repetition rate with ultrashort pulse laser, the ablation efficiency owing to high power and high repetition rate laser did not increase as much as expected due to thermal nesting effect and the shielding effect of plumes. Moreover, in some cases, thermal and secondary damage occurred at the laser irradiation part even when ultrashort pulse laser was used.

To overcome the phenomena that appear as the ultrashort pulse laser achieve high output and high repetition rate, attempts to increase the processing efficiency and productivity using ultrashort pulse laser of high output and high repetition rate by applying a technique to change the optical system setup, such as beam splitting, has drawn attention and is being investigated by various researchers. Studies to apply them to the processing of polymers that are frequently used in flexible devices are also underway.

However, techniques to change the optical system configuration such as beam splitting have limitations because it increases the complexity of the optical system and makes it difficult to align the optical system. Furthermore, the techniques to change the optical system setup simply lower the output from one beam by splitting beams or changing the beam shape, and still have the possibility that the above-mentioned effects will appear again when the output and pulse repetition rate of one beam increase.

To lower the reduction of ablation efficiency—caused by shielding effect caused by plumes generated when processing with ultrashort pulse laser of high output and high repetition rate—we propose a method of improving ablation efficiency by inducing an efficient escape route of plumes generated by laser such as increasing the welding efficiency by key hole opening, which is often used in laser welding. To induce the efficient escape route of plumes generated during laser processing, we generated a fine curvature on polymers as a preparation before processing them using ultrashort pulse laser. Then the laser was irradiated in the condition of a fine curvature. This fine curvature of polymers generated in the preliminary preparation induced an opening such as an artificial chimney along the route of the incident beam during laser processing, and enabled effective removal of plumes. To examine the hypothesis that the effective removal of plumes will improve ablation efficiency during processing using ultrashort pulse laser, we conducted research while changing the laser parameters and the fine curvature generated on polymers.

## 2. Experiment

For the ablation efficiency improvement experiment using the fine curvature of polymers in the preliminary preparation proposed in this study, we used the Nd:YVO4 laser (Time bandwidth) with a wavelength of 355 nm and a pulse width of 10 picoseconds. The laser power can be precisely controlled through a general attenuator, and the pulse repetition rate was fixed to 100 kHz, which can be operated at the maximum output, by referring to the specifications of the laser manufacturer. The laser power parameter was set to 1 W (10 μJ) and 1.5 W (15 μJ) for experiment, considering the output stability for laser oscillation.

The scan speed for the experiment was set to 100 and 150 mm/s, which allows testing with a constant velocity section considering the maximum speed that can be driven in the stage we use and the acceleration and deceleration of the stage. In addition, the experiment using the F-theta lens was conducted with the same speed for trend comparison with the stage result.

The optical system for laser processing was configured in two types as follows. First, an optical system based on objective lens(Mitutoyo), which is generally used in micro-precision machining and an optical system based on F-theta lens(SCANLAB), which uses x and y Galvano mirrors(SCANLAB). The Galvano mirror and F-theta lens are generally characterized by the possibility of high-speed scanning processing.

The system for laser processing was configured as shown in Figure 1a. The system that we used is composed of an optical system based on objective lens and an optical system based on F-theta lens. The configuration can be changed to an optical system of objective lens or F-theta lens through a flip mirror that is on the optical path.

In this experiment, for configuration based on an objective lens, we used 10× magnification of the objective lens (Mitutoyo), 0.28 N.A. and 20 mm focal length. For configuration based on based on an F-theta lens, we used a lens of 20 mm focal length. When calculating the beam waist diameter on the upper surface of the sample by the following formula, it is calculated as 1.5 μm for the objective lens and 15 μm for the F-theta lens(SCANLAB), respectively.
(1)$$d=\frac{4M^{2}\lambda f}{\pi D}$$
where *d* is the diameter of the laser beam on the focal plane, *M^2^* is the beam quality factor, λ is the wavelength, *f* is the focal distance of the focal lens, and *D* is the diameter of the laser beam incident on the lens.

Most flexible devices consist of three layers: a barrier layer, a working layer, and a substrate. In this study, multi-layer polymer samples were prepared with three layers of flexible devices, and the types and arrangement order of the polymers used in each layer are as follows; polyethylene terephthalate (PET) film having a thickness of 85 μm was employed both as the barrier layer and as the substrate while a working layer was made using a polyimide (PI) film having a thickness of 25 μm. Adhesive is applied for bonding between each layer. The samples were fabricated with a total thickness of 310 μm, a length of 50 mm, and a width of 20 mm. A schematic diagram of the structure of the fabricated multi-layer polymer sample is shown in Figure 1b. Before the experiment, the samples were washed using ethanol for 1 min to remove foreign substances such as dust on the surface. The method of improving ablation efficiency for efficient removal of plumes used in this study is to generate a fine curvature by pushing the sample from one side for a certain distance using a jig. The preparations for this experiment and the experimental setup are shown in Figure 1c.

We fabricated two jigs for this study using a 3D printer. One jig was fixed and the other jig was placed on the stage fitted with a micro-gauge. The push length, which is a control parameter, was adjusted from 0.1 to 1 mm through the micro-gauge (Mitutoyo). Once a fine curvature was generated on the sample through the preliminary preparation, we selected the surface of the sample center which is the highest position of the curvature as the focal position for laser processing. The experiments were performed using the conventional and new laser ablation methods with 1, 5, and 10 scans. After the experiment, we measured the surface condition and ablation depth of the processed sample using an optical microscope (Olympus) and FE-SEM (Hitachi) equipment.

## 3. Results

We conducted an experiment to compare the new laser ablation method proposed under the hypothesis that ablation efficiency can be improved by efficient removal of plumes in this study with the conventional laser ablation method. In this experiment, we processed the samples using a fine curvature through a preliminary preparation in an objective lens-based optical system that is used for micro precision laser processing, then we used the conventional laser ablation method and compared the results.

The fine curvature of the sample was generated by pushing the sample by 1 mm from one side with a jig through preliminary preparation of the objective lens-based optical system. The experiments were performed using the conventional and new laser ablation methods with 1, 5, and 10 scans. The experiment results using the objective lens-based optical system are shown in Figure 2a shows the cross-sections of samples observed using an optical microscope Figure 2b shows FE-SEM images of the surfaces of the samples scanned five times using the two processing methods, and Figure 2c shows graphic representation of the processed depths of the samples measured under various conditions.

As shown in Figure 2a, deeper ablation was observed in the result of processing that generated a fine curvature by pushing the sample by 1 mm in the preliminary preparation regardless of the scan speed and the irradiated laser energy condition than in the result of the conventional laser ablation method except for the result of one scan. In the FE-SEM images of surface processing conditions for the sample laser-processed with a fine curvature generated by pushing the sample by 1 mm in the preliminary preparation and the sample laser-processed by conventional laser ablation in Figure 2b, it is difficult to observe significant differences between the two processing methods. This result indicates that even though the proposed processing method using a fine curvature generated by pushing the sample by 1 mm in the preliminary preparation generates a greater ablation, no significant differences are observed in terms of the surface processing condition.

In Figure 2c, graphs of ablation depths measured under various conditions are shown, one scan did not generate a significant difference between the laser processing with a fine curvature of the sample generated by pushing the sample by 1 mm in the preliminary preparation and the conventional laser ablation. However, five or more scans improved the ablation efficiency as shown by the greater ablation depth of the laser processing with a fine curvature of the sample generated by pushing the sample by 1 mm in the preliminary preparation than the conventional laser ablation. In the graphs, yellow boxes indicate the experiment result by the conventional laser ablation method, the boxes with oblique lines indicate the experiment result using a fine curvature of the sample generated by pushing the sample by 1 mm in the preliminary preparation, and the boxes with wavy dashed lines indicate the case where the sample was cut for the full thickness of the sample used in this experiment.

The ablation efficiency was compared between the conventional laser ablation method and the proposed laser ablation method with a fine curvature of the sample generated by the preliminary preparation in F-theta lens-based optical system setup with x and y Galvano-mirrors which can perform high-speed scanning laser processing.

In the F-theta lens-based optical system setup, a fine curvature was generated by pushing the sample by 0.1, 0.5, and 1 mm using a jig through preliminary preparation. Figure 3 shows the results of the conventional laser ablation and the processing with a fine curvature generated by pushing the sample in the preliminary preparation. Figure 3a shows the cross-section observed with an optical microscope, Figure 3b shows the FE-SEM images of the surface conditions of the samples processed by the two processing methods, and Figure 3c shows the graphs of ablation depth measurement results of the samples processed under various conditions.

The results of the experiment using x and y Galvano-mirrors and F-theta lens-based optical system setup also showed that the processing with a fine curvature generated by pushing the sample in the preliminary preparation generated a deeper ablation than that of the conventional laser ablation regardless of the laser power and scan speed except for one scan. Furthermore, when the push length (curvature of the sample) was changed, a greater push length tended to generate a deeper ablation, as shown in Figure 3a. When the surface processing condition is compared using FE-SEM between the conventional laser ablation and the laser processing that generates a fine curvature by pushing the sample using a jig through preliminary preparation, it is difficult to observe significant differences, as shown in Figure 3b. When we observe the graphs of ablation depth measurements of the samples processed under various conditions in Figure 3c, the ablation depth did not show significant differences after one scan. However, when the number of scans increased, a deeper ablation was processed by the preliminary preparation that generated a fine curvature by pushing the sample using a jig than that of the conventional method. In other words, a trend of improving ablation efficiency can be observed. In the graphs, yellow boxes indicate the experiment result by the conventional laser ablation method, the boxes with oblique lines indicate the experiment result using a fine curvature of the sample generated by pushing the sample by 1 mm in the preliminary preparation, and the boxes with wavy dashed lines indicate the case where the sample was cut for the full thickness of the sample (310 μm) used in this experiment.

Furthermore, when we compare Figure 2, which is the result of the objective lens-based optical system setup, and Figure 3, which is the result of the F-theta lens-based optical system setup, the latter shows a greater ablation depth than the former regardless of the processing method by the preliminary preparation.

Figure 4 shows the changes in the fine curvature before scan and after scan according to the number of laser scans using the new laser ablation method that applies a fine curvature by pushing the sample using a jig through preliminary preparation. In this result, we can observe an interesting phenomenon.

First, the images before scanning of the objective lens-based optical system Figure 4a and the F-theta lens-based optical system Figure 4b when the sample was pushed by 1 mm using a jig show that a considerable curvature was already formed. In addition, when the curvature shape changes according to the number of laser scans are compared, the F-theta lens-based system shows a greater change in the curvature shape according to the number of scans than that of the objective lens-based system. In the objective lens-based optical system, the curvature changes to a slightly different shape from the initial curvature after five laser scans. However, in the F-theta lens-based optical system, a change in the curvature shape was observed from three scans and the curvature changed fully to a triangular shape after five scans. Thus, a highly distinct difference is observed depending on the optical system setup.

In addition, when the sample was pushed by 0.1 mm (Figure 4d) and 0.5 mm (Figure 4c) using a jig in the F-theta lens-based system, a comparison with the image before laser scan indicates that almost no curvature was generated compared to the case of pushing the sample by 1 mm. Furthermore, unlike the result of pushing the sample by 1mm, no change in the curvature shape by the number of laser scans was observed.

Figure 5 shows the images observed using an optical microscope for the surfaces of samples processed by generating a fine curvature by pushing the sample in the preliminary preparation and the samples processed by the conventional method. In this figure, the red dashed line indicates the area where delamination of the sample occurred during laser processing. When the delaminated areas are compared with the images in Figure 4, it can be seen that the delamination is observed when a large curvature of the sample is generated due to a long push length and the curvature shape changes to a triangle after gradually changing according to the number of laser scans. Furthermore, when delamination occurs, variations in the gaps between multiple layers of polymers occur in the sample. When these samples are measured using a reflective optical microscope, the reflections on the surface increase and the delamination area appears brighter. When the sample is pushed by a smaller distance (when the curvature is almost flat), no delamination appears in the results of the F-theta lens-based system as with the conventional laser ablation method.

## 4. Discussion

This study proposes a method for reducing a decreasing ablation efficiency due to a shielding effect appearing by plume which occurs during laser processing as an ultrashort pulse laser with a higher output and repetition rate is developed. The proposed method was used to increase laser ablation efficiency by generating a fine curvature on polymers as preliminary preparation to induce an efficient escape route of plumes. For experiments based on this method, two types of optical system setup were configured in this study, of which one is a scanning method utilizing a stage based on objective lens, and the other is a scanning and processing method using x-y Galvano-mirrors and F-theta lenses. Regardless of the configuration of the optical system used in the experiment, it was observed that the ablation efficiency was improved in the processing by the method proposed. The experimental results through the configuration of the optical system setup based on objective lens are shown in Figure 2 and Figure 3 shows experimental results based on the configuration of the optical system using x-y Galvano-mirrors and F-theta lenses.

Figure 2 and Figure 3 have shown the tendency of ablation depth increasing due to the fine curvature generated in the preliminary preparation. This study continues to investigate the ablation depth per number of scans due to the fine curvature generated in the preliminary preparation, as well as the efficiency improved by the new processing method in comparison to the conventional processing method. Table 1 shows the related details. Here, Table 1 shows the samples processed five scans, except for 10 scans in which the applied polymer sample is completely cut, as well as one scan in which little improvement of ablation efficiency is observed due to the small amount of ablation generated during laser processing. Moreover, the degree of curvature that is generated in the preliminary preparation was represented by the distance through which the sample is pushed. The radius of curvature generated before laser processing was measured using the side view images of sample (as shown in the first column in Figure 4) in conjunction with image analysis software (Image J).

In this case, the radiuses of curvature corresponding to the pushed distances of 0.1, 0.5, and 1 mm are 769 (769 R), 147 (147 R), and 42 mm (42 R), respectively. The results of the ablation depth per scan count indicated in Table 1 show that the ablation depth tends to decrease as the scan speed increases in the same fluence. The results suggest that the irradiated fluence per unit time decreases because the number of overlapping pulses per unit time decreases. As the radius of curvature increases, the ablation depth per scan tends to decrease. Table 1b shows the result of calculating the ablation efficiency improved in comparison to conventional processing (with no preliminary preparation). A comparison with the conventional laser ablation method for the radius of curvature of 42 mm where the push length is large (1 mm push), the ablation efficiency improved by 41% at the minimum to 69% at the maximum. Furthermore, the ablation efficiency showed a decreasing tendency with the increasing radius of curvature. However, surprisingly, an improvement of the ablation efficiency by approximately 20% was observed when compared with the conventional laser ablation method even in the condition of a very large radius of curvature of 769 mm (almost flat surface).

As previously described, the ablation efficiency has been improved depending on the application of a fine curvature generated in the preliminary preparation. Although the ablation efficiency was improved regardless of the optical system setup, positive results were observed where no significant difference on the processed surface. More surprisingly, the radius of curvature improved by approximately 20% in comparison to the conventional processing method, even at the largest 769 mm under our experimental conditions (in the near-plane status of samples). This study has compared the photographic images of plumes generated during laser processing, depending on the fine curvature generated in the preliminary preparation, and further discussed the processing method by the fine curvature generated in the preliminary preparation, as well as the mechanism of ablation improved in comparison to conventional processing methods.

First, Figure 6 shows a photograph comparing plumes according to the fine curvature generated in the preliminary preparation. Here, a shows the result by the conventional processing method, and Figure 6b shows the result by the processing method which has generated the fine curvature. The observed scanning processing conditions include the fluence of 5.7 J/cm^2^, the scan speed of 100 mm/s, and use of F-theta lens, and the radius of curvature was 42 mm (42 R). The results clearly show that plume was smoothly removed in the processing method that has generated fine curvature in comparison to the conventional processing method of Figure 6a. As shown in a, the brightest spot in the center is the interaction (processing) region where the laser was irradiated onto the polymer sample. This region shows that finely generated plumes are removed from the sample, upwards from the brightest spot, indicating that nearly generated plumes were trapped and interrupted in the removal process. However, Figure 6b, which is the result of the process generating a fine curvature, shows that the plume generated during laser processing is smoothly removed from the sample with a certain directionality. As proposed in this study, the result suggests that the plume generated during laser processing is removed smoothly through the chimney-like opening.

The mechanism of improvement in ablation efficiency by the proposed laser ablation method using a fine curvature generated by pushing the sample with a jig through preliminary preparation can be explained using Figure 7a shows the formation of plumes by the conventional laser ablation. Figure 7b shows the preliminary preparation in which plumes are processed by laser processing with a fine curvature generated by pushing the sample with a jig. Here, the number of scans increases to the right. As shown in this figure, the amount of plumes increases with the number of scans. In the case of the laser ablation method using a fine curvature of the sample generated through preliminary preparation, more openings are generated in the incident direction of laser beam during the laser processing, and plumes are removed efficiently upwards like a chimney. When this efficient removal of plumes occurs, the ratio of laser beam transmitted to the inside the material increases, thus increasing the ablation depth and improving the ablation efficiency as a result.

However, as shown in Figure 7a, the conventional laser ablation (flat ablation) does not efficiently remove plumes generated, in other words, openings are not formed smoothly. Consequently, the plumes block the path of laser beam transmitted to the inside of the material, and most laser energy is scattered by and absorbed in the plumes, lowering the ratio of direct absorption by the material. This process results in lower ablation efficiency. Therefore, the experiment results in this study confirmed that efficient removal of plumes enhances the laser ablation efficiency as stated in our hypothesis.

This study continues to once again compare the change in curvatures and shapes of the samples before and after laser processing through the experimental results with the F-theta lens-based optical system setup with varied curvatures. Figure 8 shows the conditions where the radii of curvature before processing, are 42, 147, and 769 mm, respectively, as well as the change in radii of curvature and shapes after laser processing. The left side of Figure 8 refers to the status before laser processing, and its right side indicates the status after laser processing. In Figure 8 the first row corresponds to the cases with a radius of curvature of 42 mm before processing, the second row corresponds to a radius of curvature of 147 mm, and the third row corresponds to a radius of curvature of 769 nm. Figure 8a indicates the experimental results of cases having the smallest radius of curvature among the experimental conditions of this study. The image on the left side, which was obtained before laser processing, shows that a considerable degree of curvature has been formed in the sample in the preliminary preparation, whose radius of curvature was measured as approximately 42 mm by using Image J software. The image on the right side after laser scanning under these conditions shows that the arch shape of the semicircle before laser processing has been changed to triangular shape after laser scanning. Figure 8a having the highest degree of curvature, indicated the highest difference as approximately 30% of the improvement in ablation efficiency. However, Figure 5, which corresponds to the results of observing the experiment with various degrees of curvature and optical system using an optical microscope, shows the delamination phenomenon. This phenomenon is discussed below in more detail.

In Figure 8b,c which show the curvature before laser processing, a relatively lower degree of curvature has been formed in comparison to Figure 8a. In particular, Figure 8c shows the near-plane curvature. In Figure 5, which corresponds to the results of observing the experiment with various degrees of curvature and optical system using an optical microscope, shows no occurrence of delamination phenomenon. Interestingly, in terms of improvement in ablation efficiency, Figure 8c having the near-plane curvature shows improvement in ablation efficiency by approximately 20%, which suggests that the efficient removal phenomenon of the plume still occurs.

As previously described, we discussed changes in curvature and shapes under varied curvature conditions before and after laser scanning. This study continues to describe the reasons for the occurrence of the delamination phenomenon in processing with a high degree of curvature.

Figure 8, which corresponds to the results of observing the change in curvature and shapes under various curvature conditions before and after laser scanning, shows that the arch curvature of the circle before laser processing has been changed to the triangular shape after laser scanning. In the case of samples having changed shapes, Figure 5, which corresponds to the results of observing the experiment with various degrees of curvature using an optical microscope, shows the delamination phenomenon. Figure 9 shows the reasons for occurrence of delamination only when the curvature is high. Figure 9 described the phenomenon caused by laser scanning in the case of high and low degrees of curvature. Here, Figure 9a shows the case where the curvature is low, and Figure 9b shows the case where the curvature is high. Comparing inward force and outward force applied before the laser scanning process in Figure 9a,b, inward force and outward force are more likely to be higher at the neutral axis when the applied curvature is high in comparison to the case with low curvature. In the case of applying low curvature having relatively small inward force and outward force, the difference in inward force and outward force occurred along the neutral axis is insignificant, which is no changed depending on laser scanning processing. Thus, the adhesion of the adhesive layer could be sufficiently adhered to the top layer. On the other hand, in Figure 9b having a high curvature, inward force, and outward force are more likely to be higher at the neutral axis before laser scanning processing. Since a relatively large amount of inward and outward forces are applied along the neutral axis before laser scanning, it is presumed that delamination occurs because the adhesive strength of the adhesive layer exceeds the range that can sufficiently adhere to the uppermost layer during laser scanning.

The contents mentioned in the previous paragraph have been discussed in relation to the delamination between polymer layers. We will start discussion about plastic deformation and mechanical fracture that might occur during laser processing in our proposed method. In our previous study, we conducted the stress/strain analysis on the UV picosecond laser (which is the same laser employed in the current study) induced grooves on PET film while the bending is applied, using the finite element model [31]. In summary, when the 100um PET thin film is bending after laser scribing on the surface, the stress is concentrated at the tip of the laser induced groove. The analysis also suggests that if the high stress zone undergoes plastic deformation, each deformation zone was very small (extending less than 10 μm from the tip along plate axis) and the maximum strain at the tip was smaller than the strain at the fracture. We also proposed that the thin film can be bent up to a 0.38 mm radius of curvature without experiencing mechanical failure after the previously mentioned laser scribing. Considering the previous result, the chance for plastic deformation taking place is very small in the current study. It should be noted that the radius of curvature which is tested in the current study is about (at least) 100 times larger than the optimal radius of curvature proposed in the previous study. Which means the laser induced groove in the current study is much less susceptible to plastic deformation. In addition to the analytical study point from our previous work, it should be noted that the central focus of the current study is to propose a better ablation condition for the full cutting. If plastic deformation exists at the tip of the groove, it does not impact on the fundamental result that the film is fully cut.

## 5. Conclusions

We fabricated multiple layers of polymers, which have flexibility, and used them in this experiment. Most flexible devices are composed of multiple layers of polymers and this experiment was configured with similar conditions. We proposed a method of improving laser ablation efficiency by efficient removal of plumes generated during laser processing. This method is to generate a fine curvature on the sample through preliminary preparation before laser processing. In this experiment, we used ultrashort pulse laser of the UV wavelength with a pulse width of 10 picoseconds and a pulse repetition rate of 100 kHz. The verification experiment was performed using the laser energy of 10 and 15 μJ and the scan speeds of 100 and 150 mm/s. The jig for preliminary preparation was fabricated using a 3D printer. Two optical system setups were used in this experiment: the objective lens-based optical system setup, which is often used in micro-precision processing, and the F-theta lens-based optical system setup with x and y Galvano-mirrors.

The laser processing results showed a tendency of improving ablation efficiency when the samples were processed with a fine curvature generated through preliminary preparation regardless of the optical system setup, laser power, and scan speed except the case of one scan. Furthermore, no significant differences between the conventional laser ablation and the ablation using a fine curvature on the sample were found when the sample surfaces were observed using FE-SEM after laser scanning. When an experiment was performed with different curvatures, a large curvature (large push length of the sample), the ablation efficiency improved significantly, but unwanted delamination phenomenon was observed. The delamination phenomenon did not occur when the curvature was small (small push length). An interesting result of this study is that the ablation efficiency improved by approximately 20% even when the radius of curvature was not much different from the plane. The result of the enhancement ablation efficiency is that the plume generated during processing is trapped in the conventional processing, whereas in the case of the proposed processing, the generated plume is efficiently removed. This difference is because, in the case of the proposed processing, opening occurs like a chimney due to the fine curvature generated through the preliminary preparation process. The proposed method is expected to be applicable to flexible devices such as flexible displays and flexible batteries. Since the ablation efficiency showed improvement even with a very small curvature, the proposed method could be applied to many other fields if the method of implementing the curvature is improved.

## Figures and Tables

**Figure 1 polymers-12-00959-f001:**
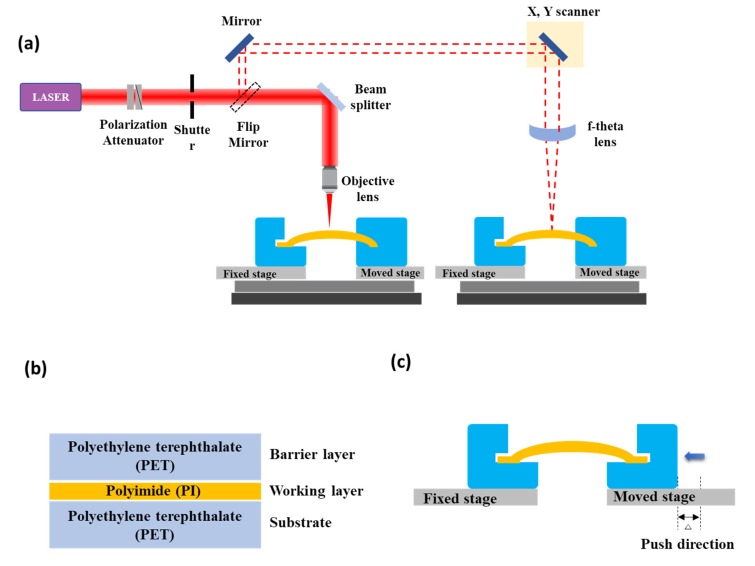
(**a**) Overview of laser processing system in which an objective lens-based and F-theta lens-based optical systems (**b**) schematic diagram of multi-layer polymer sample (Simplified flexible device) (**c**) Overview of the experiment method that generates a fine curvature as a preliminary preparation.

**Figure 2 polymers-12-00959-f002:**
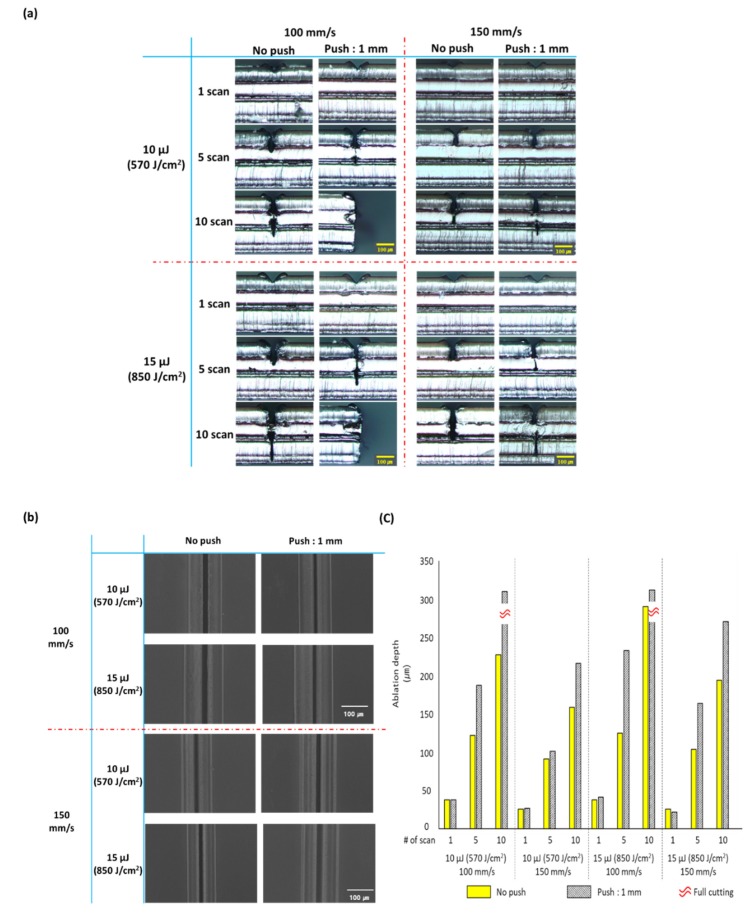
Processing result of the fine curvature on the sample generated in the preliminary preparation in an optical system based on an objective lens in comparison with the conventional laser ablation result: (**a**) cross-section measurement result using an optical microscope, (**b**) comparison of the sample surface condition by FE-SEM between the conventional laser ablation and the processing by the preliminary preparation after five scans, (**c**) measurements of ablation depth through the conventional laser ablation and the processing by the preliminary preparation under various conditions.

**Figure 3 polymers-12-00959-f003:**
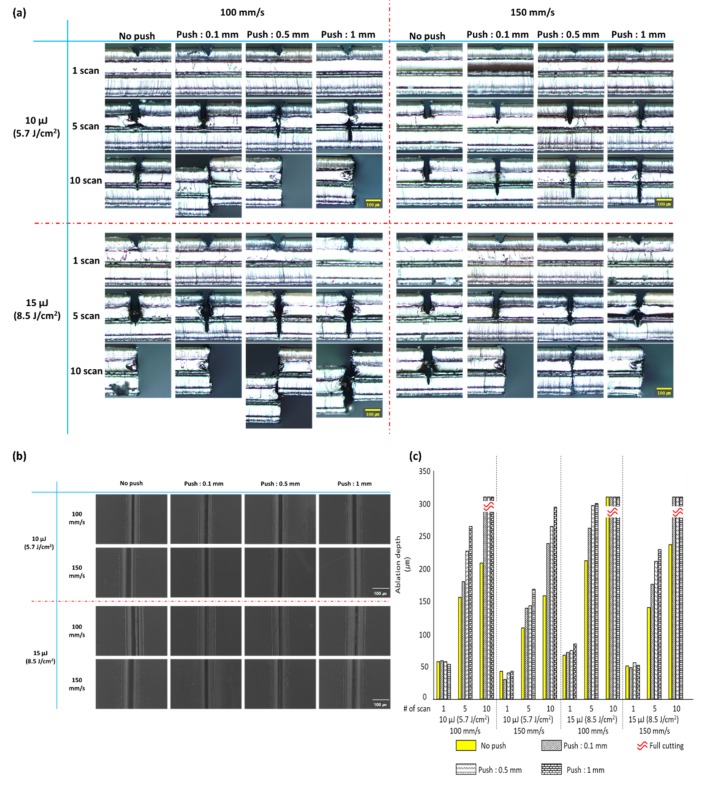
Comparison of the processing results between the conventional laser ablation method and the processing method using a fine curvature generated on the sample in the preliminary preparation in the F-theta lens-based optical system: (**a**) cross-section measurement result observed with an optical microscope, (**b**) FE-SEM images of the surface conditions of the samples processed by the conventional method and by the preliminary preparation after five scans, (**c**) ablation depth measurement results of processing by the conventional method and by the preliminary preparation.

**Figure 4 polymers-12-00959-f004:**
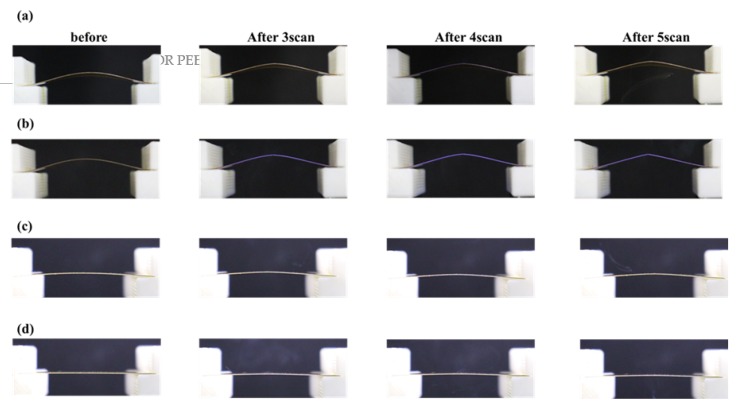
Images of the samples applying a fine curvature by pushing the sample in the preliminary preparation before laser processing and after scans: (**a**) with objective lens, (**b**) F-theta lens (push length: 1 mm), (**c**) F-theta lens (push length: 0.5 mm), (**d**) F-theta lens (push length: 0.1 mm).

**Figure 5 polymers-12-00959-f005:**
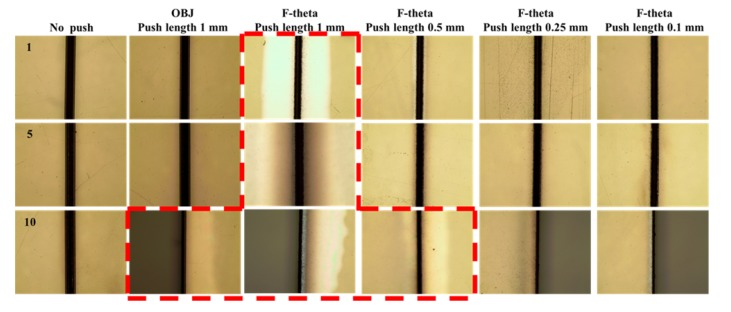
Top surface images of the samples processed by the conventional method and by the new laser ablation method observed with a reflective optical microscope at 10 μJ and 100 mm/s (red dash line: delamination occurrence area).

**Figure 6 polymers-12-00959-f006:**
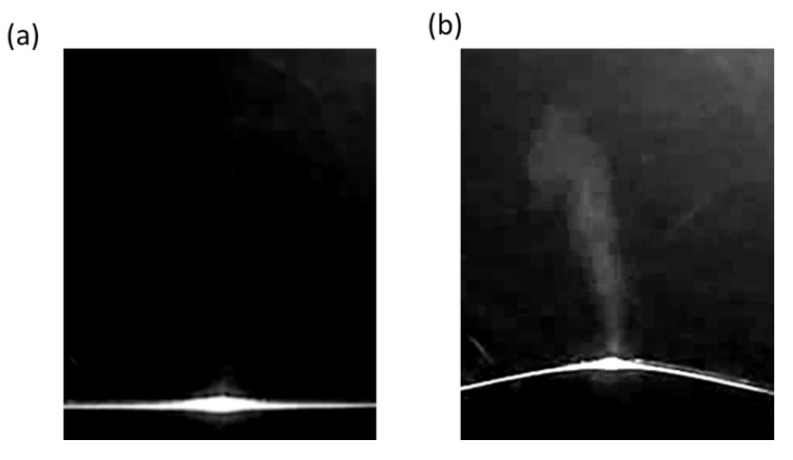
Comparison of plumes depending on whether or not the fine curvature is generated through the preliminary preparation process (**a**) conventional processing method, (**b**) processing method by micro curvature at 5.7 J/cm^2^ 100 mm/s with F-theta lens 42 R.

**Figure 7 polymers-12-00959-f007:**
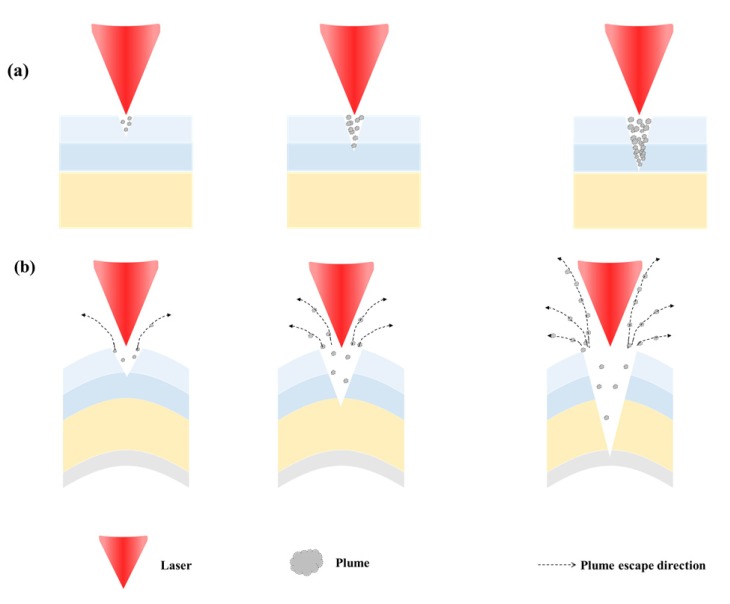
Enhancement of laser ablation by curvature of polymer produced through preliminary preparation (exhaust of plume produced) (**a**) conventional laser ablation (**b**) proposed method.

**Figure 8 polymers-12-00959-f008:**
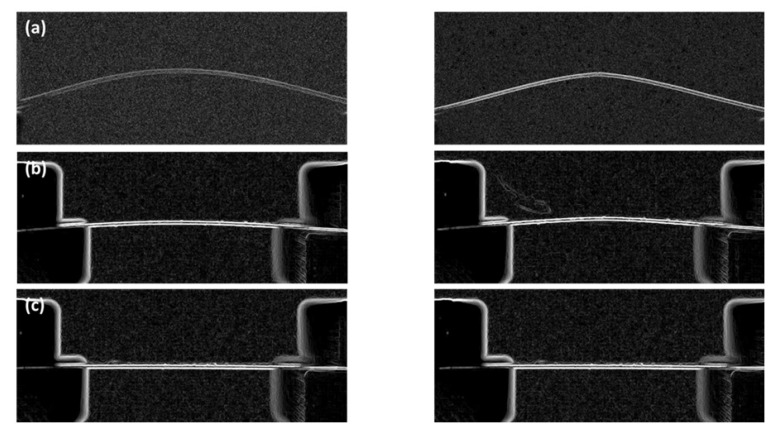
Curvature comparison image of polymer samples before (left) and after (right) laser scanning after image processing. (**a**) Radius of curvature 42 mm by F-theta lens, (**b**) radius of curvature 147 mm by F-theta lens, (**c**) radius of curvature 769 mm by F-theta lens.

**Figure 9 polymers-12-00959-f009:**
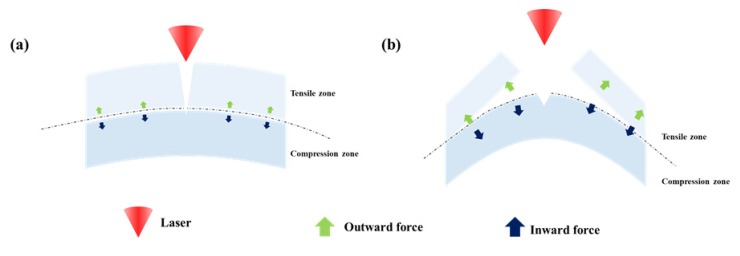
Process of delamination phenomenon during processing according to the degree of curvature of polymer (**a**) smaller curvature (no delamination), (**b**) bigger curvature.

**Table 1 polymers-12-00959-t001:** Comparison of ablation depth (**a**) and ablation efficiency improvement ratio in comparison with the flat surface (**b**) after five scans under various conditions between the conventional laser ablation and the preliminary preparation.

(a)						(b)				
Flunce (J/cm^2^)	Scan Speed (mm/s)	Ablation Depth (μm/scan)				Flunce (J/cm^2^)	Scan Speed (mm/s)	Ablation Efficiency Improvement Ration in Comparision with the Flat Surface (%)		
		∞	769 R	147 R	42 R			769 R	147 R	42 R
5.7	100	31.3	36	45.3	52.9	5.7	100	**15**	**45**	**69**
	150	21.9	27.9	28.7	33.6		150	**27**	**31**	**54**
8.5	100	42.5	52.4	59.4	59.9	8.5	100	**23**	**40**	**41**
	150	28.1	35.2	42.2	45.9		150	**25**	**50**	**63**

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
