# Peer review of "Development of Method Enhanced Laser Ablation Efficiency According to Fine Curvature of the Polymer through the Preliminary Preparation Process Using UV Picosecond Laser"

_polymers, 2020, doi:10.3390/polym12040959_

Round 1

Reviewer 1 Report

The present paper investigates the possibility of enhancing the laser ablation efficiency by the introduction of a fine curvature of plastic samples, which allows for a more efficient removal of plumes generated during processing. The developed approach is very simple, and according to the finding of the authors, allows for a more efficient ablation of material. However, in my opinion, the novelty and the scientific soundness of the paper are not up to the standards of the journal. This does not seem a true scientific paper, but rather a good technical report. I would suggest publication on a more specific journal.

Here is a list of the main issues of the paper:

  • No detail is given about the type of plastic samples used for the experiment. Is it a rigid, or a rubbery plastic? This is very important, since the proof of concept is given for only one type of plastic, and there is no evidence that it could work with other types of plastic.
  • No detail is given about the effect of curvature on the final shape of the part. Also in view on the proposed ablation mechanism, and the shape change to triangular shape, it is not clear if there is the possibility that some plastic deformation remains on the sample after ablation. This would require a more detailed viscoelastic-analysis of the plastic .
  • Discussion section is very similar to results section, with the exception that radius of curvature is used instead of jig displacement
  • The change of the shape from curvilinear to triangular is not so strange. It mainly results from the thickness reduction in the ablation region, which causes the curvature to be concentrated in a very narrow region close to it.

Author Response

Thank you very much for all the comments from the reviewers.  We put our best efforts to address all the issues raised by each reviewer.  We also made the appropriate changes in the manuscript where they are needed.

Point 1: No detail is given about the type of plastic samples used for the experiment. Is it a rigid, or a rubbery plastic? This is very important, since the proof of concept is given for only one type of plastic, and there is no evidence that it could work with other types of plastic.

Response 1: 

We added detailed information on the thin film sample test in the current study.  The detailed information is as follows:

Most flexible devices consist of three layers: a barrier layer, a working layer, and a substrate. In this study, multi-layer polymer samples were prepared with three layers of flexible devices, and the types and arrangement order of the polymers used in each layer are as follows; polyethylene terephthalate (PET) film having a thickness of 85㎛was employed both as the barrier layer and as the substrate while a working layer was made using a polyimide (PI) film having a thickness of 25㎛.  Adhesive is applied for bonding between each layer.

We have added the detailed information on the polymer sample in the experimental section.

Point 2: No detail is given about the effect of curvature on the final shape of the part. Also in view on the proposed ablation mechanism, and the shape change to triangular shape, it is not clear if there is the possibility that some plastic deformation remains on the sample after ablation. This would require a more detailed viscoelastic-analysis of the plastic.

Response 2:  As the reviewer pointed out, the deformation of the polymer film during /after the laser ablation is an important issue.  

In our previous study, we conducted the stress/strain analysis on the UV picosecond laser (which is the same laser employed in the current study) induced grooves on PET film while the bending is applied, using the finite element model. [32]

The typical outcome of the previous result is introduced in the following figure

The digital images of the films obtained after the bending test (top) and the simulation result of Von Mises stress distribution, after a bending test of laser-induced grooves in the material by the finite element analysis (bottom).

In summary, when the 100um PET thin film is bending after laser scribing on the surface, the stress is concentrated at the tip of the laser induced groove. The analysis also suggests that if the high stress zone undergoes plastic deformation, each deformation zone was very small (extending less than 10um from the tip along plate axis) and the maximum strain at the tip was smaller than the strain at the fracture. We also proposed that the thin film can be bent up to a 0.38mm radius of curvature without experiencing mechanical failure after the previously mentioned laser scribing.

Considering the previous result, the chance for plastic deformation taking place is very small in the current study.  It should be noted that the radius of curvature which is tested in the current study is about (at least) 100 times larger than the optimal radius of curvature proposed in the previous study. Which means the laser induce grooved in the current study is much less susceptible to plastic deformation.

In addition to the analytical study point from our previous work, it should be noted that the central focus of the current study is to propose a better ablation condition for the full cutting. If plastic deformation exists at the tip of the groove, it does not impact on the fundamental result that the film is fully cut.

We have added the issue of plastic deformation, to the discussion section in the last paragraph.

Point 3: Discussion section is very similar to results section, with the exception that radius of curvature is used instead of jig displacement

Response 3:  

In order to address point 3, we have added the issue of plastic deformation, as outlined in the second point, to the discussion section in the last paragraph.

Point 4: The change of the shape from curvilinear to triangular is not so strange. It mainly results from the thickness reduction in the ablation region, which causes the curvature to be concentrated in a very narrow region close to it.

Response 4:  

The reviewer makes a valid point which we agree with. We have added mention of this point into the discussion section as part of our addressing the second point.

Reviewer 2 Report

The authors proposed a processing method that can improve the ablation efficiency by providing an escape path of the plume. The ideas are new and very interesting. And the paper is well-written. I only have a few suggestions to improve the paper.

  1. From Fig. 2 and Fig. 3, it can be seen that the curvature is quite small. Can the author provide the quantitative value of the curvature? It can be measured or calculated directly. It may help readers to understand the interesting result. Just curious, can you use a blowing device to remove the generated plume during the ablation process.
  2. Several typos should be corrected, such as “in the we experimented windows”, “material surface s estimated”.

Author Response

Response to Reviewer 2 Comments

Thank you very much for all the comments from the reviewers.  We put our best efforts to address all the issues raised by each reviewer.  We also made the appropriate changes in the manuscript where they are needed.

Point 1: From Fig. 2 and Fig. 3, it can be seen that the curvature is quite small. Can the author provide the quantitative value of the curvature? It can be measured or calculated directly. It may help readers to understand the interesting result. Just curious, can you use a blowing device to remove the generated plume during the ablation process.

Response 1:

   The degree of curvature that is generated in the preliminary preparation was represented by the distance through which the sample is pushed. The radius of curvature generated before laser processing was measured using the side view images of the samples (as shown in the first column in Figure 4) in conjunction with image analysis software (Image J).

 In this case, the radiuses of curvature corresponding to the pushed distances of 0.1 mm, 0.5 mm, and 1 mm are 769 mm (769 R), 147 mm (147 R), and 42 mm (42 R), respectively.

We introduce this information into the second paragraph of the discussion section.

Yes. In general laser processing, a blowing device is also applied to remove the plume.

However, we didn’t use a blowing device in this study but we expect that the use of one would enhance plume removal during the application of our proposed method.

Point 2: Several typos should be corrected, such as “in the we experimented windows”, “material surface s estimated”.

Response 2:

As the reviewer pointed out, we have corrected the wrong phrases by removing or revising the erroneous phrases. The changes can be seen in the 5th paragraph of the experiment section and abstract.

Reviewer 3 Report

The manuscript „Development of method enhanced laser ablation efficiency according to fine curvature of the polymer through the preliminary preparation process using UV picosecond laser” proposes the uses of ultrashort pulsed laser to process some scratches in polymers. The results are interesting but improvements are required.

-In the Figure 2 and Figure 3 may be useful to give and the FESEM images in relation with optical images: 1 scan, 5 scans, 10 scans. Where the sample did not break.

-Figure 8 and Figure 9 must to be denoted as being schematic representation of…. Because are not results obtained experimental.

-Too many repetitions are in the text.

-Some paragraphs from the Results part must to be introduced in the Experimental part.

- Give more informations about the polymers preparations steps (materials, methods, …)

Author Response

Response to Reviewer 3 Comments

Thank you very much for all the comments from the reviewers.  We put our best efforts to address all the issues raised by each reviewer.  We also made the appropriate changes in the manuscript where they are needed.

Point 1: In the Figure 2 and Figure 3 may be useful to give and the FESEM images in relation with optical images: 1 scan, 5 scans, 10 scans. Where the sample did not break.

Response 1:

As the reviewer points out, the use of FE-SEM images in conjunction with optical microscope images could reveal pertinent additional details in regards to the cutting conditions. However, optical microscope images are sufficient to confirm the ablation efficiency generated by our fine curvature enhanced laser ablation process.

Instead, we used FE-SEM images to check for the presence of collateral damage on the surface in order to compare the quality of our fine curvature-enhanced cutting method to the traditional cutting method.

Point 2: Figure 8 and Figure 9 must to be denoted as being schematic representation of…. Because are not results obtained experimental.

Response 2:

The reviewer points out that Figures 8 and 9 should be denoted as being schematic representations but I suspect this is in reference to Figures 7 and 9 since Figure 8 is an experimentally obtained result. In response, we have specified that Figure 8 is an experimental result and that Figures 7 and 9 are schematic results and have clarified their representation as schematics by removing the extraneous pre-ablation portions of their images.

Point 3: Too many repetitions are in the text.

Response 3:  

In order to address point 3, we have added the issue of plastic deformation to the discussion section in the 8th paragraph as well as deleting repeated points from it. (refer to the discussion section in the 1st paragraph, and 4th paragraph)

Point 4: Some paragraphs from the Results part must to be introduced in the Experimental part

Response 4: 

As the reviewer points out there were parts in the Results section that should be moved to the experimental section. As such, the following phrase referring to the scanning number was moved from the results section and included in the experimental section.

“The experiments were performed using the conventional and new laser ablation methods with 1, 5, and 10 scans”

Point 5: Give more informations about the polymers preparations steps (materials, methods, …)

Response 5:  

We added detailed information on the thin film sample test in the current study.  The detailed information is as follows:

Most flexible devices consist of three layers: a barrier layer, a working layer, and a substrate. In this study, multi-layer polymer samples were prepared with three layers of flexible devices, and the types and arrangement order of the polymers used in each layer are as follows; polyethylene terephthalate (PET) film having a thickness of 85㎛was employed both as the barrier layer and as the substrate while a working layer was made using a polyimide (PI) film having a thickness of 25㎛.  Adhesive is applied for bonding between each layer.

We have added the detailed information on the polymer sample in the experimental section.

Round 2

Reviewer 1 Report

I havo no specific concern about the revised version of the paper. however, I must advise that the overall merit of the paper and the level of innovation are not up to the standards of the Journal, as I pointed out in my previous review.

I suggest publication on a more specific technical journal

Reviewer 3 Report

The manuscript was improved and can be published in the present form.